

# Cloud Products from the Earth Polychromatic Imaging Camera (EPIC): Algorithms and Initial Evaluation

Yuekui Yang[1], Kerry Meyer[1], Galina Wind[2,1], Yaping Zhou[3,1], Alexander Marshak[1], Steven Platnick[1], Qilong Min[4], Anthony B. Davis[5], Joanna Joiner[1], Alexander Vasilkov[2], David Duda[2,6], Wenying Su[6]

5   [1]NASA Goddard Space Flight Center, Greenbelt, MD.
[2]Science Systems and Applications Inc., Lanham, MD
[3]Morgan State University, Baltimore, MD
[4]State University of New York at Albany, Albany, NY
[5]Jet Propulsion Laboratory, California Institute of Technology, Pasadena, CA
10   [6]NASA Langley Research Center, Hampton, VA

*Correspondence to*: Yuekui Yang (yuekui.yang@nasa.gov)

**Abstract.** This paper presents the physical basis of the EPIC cloud product algorithms and an initial evaluation of their performance. Since June 2015, EPIC has been providing observations of the sunlit side of the Earth with its 10 spectral channels ranging from the UV to the near-IR. A suite of algorithms has been developed to generate the standard EPIC Level 2 Cloud Products that include cloud mask, cloud effective pressure/height, cloud optical thickness, etc. The EPIC cloud mask adopts the threshold method and utilizes multichannel observations and ratios as tests. Cloud effective pressure/height is derived with observations from the $O_2$ A-band (780 nm and 764 nm), and B-band (680 nm and 688 nm) pairs. The EPIC cloud optical thickness retrieval adopts a single channel approach where the 780 nm and 680 nm channels are used for retrievals over ocean and over land, respectively. Comparison with co-located cloud retrievals from geosynchronous earth orbit (GEO) and low earth orbit (LEO) satellites shows that the EPIC cloud product algorithms are performing well and are consistent with theoretical expectations. These products are publicly available at the Atmospheric Science Data Center at the NASA Langley Research Center for climate studies and for generating other geophysical products that require cloud properties as input.

## 1 Introduction

25   Since June 2015, the EPIC aboard the Deep Space Climate Observatory (DSCOVR) has been providing observations of the sunlit side of the Earth at the L1 Lagrangian point approximately one million miles from the Earth. The simultaneous coverage of the Earth from sunrise to sunset is a capability never available from any other spacecraft or Earth observing platform in the past. These observations provide new opportunities in climate research and applications (e.g., Marshak et al. 2018, Holdaway and Yang 2016a,b). One important contribution of EPIC is the ability to observe and retrieve key radiative

30   properties of clouds, which are of critical importance for understanding the current climate system and for predicting climate





change (e.g., Boucher et al., 2013, and references therein). The Decadal Survey for Earth Science and Applications from Space (National Academies of Sciences, Engineering, and Medicine. 2018) lists "how changing cloud cover and precipitation will affect climate, weather, and Earth's energy balance in the future" as one of the key science questions and places the observation of clouds and precipitation on the priority list. Equipped with 10 spectral channels ranging from the

UV to the near-infrared, EPIC observations provide essential information for cloud system monitoring and cloud product development.

The focal plane of the EPIC system is a 2048 x 2048 pixel CCD array. Images are obtained using a 10-color filter wheel assembly and shutter, the operation of which takes about 7 minutes to acquire a complete set of spectral observations as shown in Figure 1. The pixel size of EPIC observations is ~8km at nadir. The data from all 10 spectral channels are projected

to a common grid in the Level-1B (L1B) radiance product. A suite of algorithms has been developed for generating the standard EPIC Level 2 Cloud Products that include cloud mask, cloud effective pressure/height (CEP/CEH), cloud optical thickness (COT), etc. These algorithms use as input the observations from five EPIC channels, namely the 388 nm and the two pairs of $O_2$ A-band (780 and 764 nm) and B-band (680 and 688 nm) reference and absorption channels. These observations provide the most cloud information content and are close to each other in observation time, which is important

for reducing uncertainties resulting from temporal changes in clouds and the rotation of the earth. Table 1 gives a summary of the EPIC L2 Cloud Products. These products, which are publicly available at the Atmospheric Science Data Centre at the NASA Langley Research Center, provide cloud properties of the sunlit side of Earth and are being used in applications such as cloud screening for aerosol property retrievals, ocean color studies, trace gas retrieval corrections etc.

The remainder of the paper is organized as follows: Section 2 presents the algorithm theoretical basis for the standard

EPIC L2 Cloud Products; Section 3 presents an initial assessment of the performance of these products through comparison with results from co-located geosynchronous earth orbit (GEO) and low earth orbit (LEO) observations; a summary and discussions are presented in Section 4.

## 2 EPIC Cloud Product Algorithms

The three components of the EPIC Cloud Product system, cloud mask, CEP/CEH retrieval, and COT retrieval can be

run independently. The EPIC cloud mask serves as input to both the CEP/CEH and COT retrievals, where only pixels classified as cloudy are processed.

### 2.1 EPIC Cloud Mask Algorithm

The basis for cloud detection is the contrast between cloud and the background surface. To separate cloudy and clear pixels, satellite missions usually adopt the threshold method, which classifies a pixel through comparing the values of an

observed quantity, such as the bidirectional reflectance factor (BRF), to a predefined threshold (e.g., Ackerman et al. 2010, Yang et al. 2007, Rossow and Garder, 1993). Moreover, multiple thresholds can be defined such that the cloud mask is





assigned a confidence level. Similarly, the goal of the EPIC cloud mask is to label each pixel as clear with high confidence, clear with low confidence, cloudy with low confidence, or cloudy with high confidence. Because EPIC observations are limited to the shortwave channels, the widely used IR-based tests (e.g., temperature contrast) are not available. However, effective tests can still be constructed. To do that, the earth's surface is separated into three types: land, ocean, and snow/ice;

two tests are applied to each surface type. Table 2 lists the EPIC cloud detection tests used for each surface type.

The two tests used in the EPIC cloud mask over snow/ice-free land are the 388 nm BRF and the EPIC $O_2$ A-band ratio, i.e., $R_{764}/R_{780}$, where $R_{764}$ is the 764 nm absorption channel BRF and $R_{780}$ the 780 nm reference channel BRF.

The utility of the 388 nm channel for cloud detection is due to the fact that surface reflectance in this channel is usually small while clouds are relatively bright over snow/ice-free land (Herman et al. 2001). To accommodate a wide range of sun-

view geometries, the contribution from Rayleigh scattering is first removed from the observed BRF. A truly accurate Rayleigh correction requires knowledge of the reflecting layer height and its microphysical properties. However, for the more qualitative purposes of cloud detection, we can apply a simple Rayleigh correction based on the assumption that the observed reflectance comes from the interaction of a Lambertian surface and a Rayleigh layer above. Then the derived Lambertian-equivalent reflectivity (LER) (e.g., Herman and Celarier, 1997) can be compared with the surface albedo to

decide if clouds are present. The EPIC observed BRF at the top of atmosphere (TOA) $R_{TOA}$ can be expressed as:

$$R_{TOA} = R_R + \frac{T_R R_{LER}}{1 - S_R R_{LER}} \qquad (1)$$

where $R_{LER}$ is the Lambertian equivalent reflectance to be derived; $R_R$, $T_R$, and $S_R$ are the Rayleigh path reflectance, two-way transmittance and spherical albedo, respectively, which are calculated with the analytical solutions described in Vermote and Tanre (1997). Thresholds for the 388 nm test are based on the monthly surface reflectance climatology derived from the

Global Ozone Monitoring Experiment-2 (GOME-2) and the SCanning Imaging Absorption SpectroMeter for Atmospheric CHartographY (SCIAMACHY) missions (Tilstra et al. 2017). In practice, if the Rayleigh corrected LER is larger than the surface albedo, then the scene is labelled as cloudy; otherwise, it is clear. The uncertainty in the surface albedo provided by the dataset (Tilstra et al. 2017) is used to decide the confidence level.

The selection of the $O_2$ A-band ratio as a cloud mask test is based on the fact that everything being equal, the ratio is

higher with the presence of clouds compared to clear sky cases, because $O_2$ absorption is proportional to the air mass above the reflecting layer. Thresholds for this test are a function of surface elevation, which is based on the Global Gridded Elevation and Bathymetric (ETOPO5) dataset (Edwards, 1989). While the $O_2$ B-band ratio is also useful for cloud detection for the same reason (and is used over snow/ice as discussed below), the A-band ratio is selected for use over land because it provides better skill than that of the B-band due to its higher sensitivity to the photon path length change (Yang et al. 2013).

Over ocean, the BRFs of the 680 nm and the 780 nm channels are used for cloud detection, because clouds and the sea surface contrast well in both channels. Although the observations are highly correlated, the two channels complement each other over coastal and shallow water regions due to differences in surface reflectance. Similar to the 388 nm test over land, a



Rayleigh correction is also applied to both the 680 nm and 780 nm BRFs. Thresholds are derived based on the Cox-Munk (Cox and Munk, 1954a, b) ocean reflectivity model by assuming a fixed 6 m/s wind.

Over snow- and ice-covered regions, both the $O_2$ A- and B-band ratios are used for cloud detection. As discussed above for the A-band ratio test over land surfaces, the relative shorter photon path lengths under cloudy conditions result in higher

ratios. The thresholds are selected based on radiative transfer simulations with the Discrete Ordinates Radiative Transfer model (DISORT) (Stamnes et al., 1988). Unless otherwise mentioned, the radiative properties of the atmosphere at A- and B-band used in this paper are calculated with the Line-By-Line Radiative Transfer Model (LBLRTM) (Clough et al., 2005) using the 1976 U.S. standard atmosphere and monochromatic radiative transfer results were convoluted with the filter functions. Detailed properties of the EPIC A- and B-band channels are described in Yang et al. (2013).

The procedure of generating the EPIC cloud mask include two steps. First, for a given surface type, each of the two tests leads to an independent cloud mask represented with an integer value:

$CloudMask_{i=1,2}$ = {1: Clear with High Confidence; 2: Clear with Low Confidence;

3: Cloudy with Low Confidence; 4: Cloudy with High Confidence} (2)

Second, the final cloud mask is assigned based on the sum of the individual tests, such that:

$\sum_{i=1}^{2} CloudMask_i$ = { ≤ 3: Clear with High Confidence; 4: Clear with Low Confidence;

5 and 6: Cloudy with Low Confidence; ≥7: Cloudy with High Confidence} (3)

Figure 2 shows an example of the EPIC cloud mask for the observations at 08 UTC on Aug. 18, 2016. As can be seen from the figure, the EPIC cloud mask (Figure 2b) matches the corresponding RGB (Figure 2a) well. The fraction of the four scene types, i. e. clear with high confidence (ClrHC), clear with low confidence (ClrLC), cloudy with low confidence

(ClcLC), and cloudy with high confidence (CldHC), are 33.1%, 3.6%, 1.4% and 61.9%, respectively. From the cloud mask, it is straightforward to derive Earth's daytime cloud coverage (Figure 2c); for this granule, the total cloud coverage, including cloudy with low and high confidence, is 63.3%.

**2.2 EPIC CEP/CEH Algorithm**

The EPIC CEP/CEH is derived with from observations in the $O_2$ A-band (780 nm and 764 nm), and B-band (680 nm

and 688 nm) pairs. The idea of using $O_2$ absorption for cloud height retrieval has been investigated extensively and has been implemented in several operational algorithms (e.g., Ferlay et al. 2010, Davis et al., 2009, Vasilkov et al., 2008, Wang et al., 2008, Lindstrot et al., 2006, Kokhanovsky et al., 2005, Min et al., 2004, Vanbauce et al., 2003, Daniel et al., 2003, Koelemeijer et al., 2001, Stephens and Heidinger, 2000, Buriez et al., 1997). With the $O_2$ A- and B-band observations, EPIC provides an unprecedented opportunity to monitor cloud height from the L1 point. An information content analysis on EPIC

A- and B-band observations is provided in Davis et al. (2018a, b).



Figure 3 shows the sensitivity of the EPIC A- and B-bands to the reflecting layer height change, represented here by the derivative of the above-cloud two-way transmittance (T) with respect to height (z). While the A-band has stronger absorption than B-band. The sensitivity to cloud height is clear for both channels. For example, perturbing by 1 km the altitude of a reflecting layer at 5 km changes the atmosphere two-way transmittance by ~8% for the A-band and ~4% for the B-band.

Since oxygen is a well-mixed gas, it would be easy to derive cloud top height from radiance measurements in the absorption band if the cloud behaved optically like a hard target. However, multiple scattering along the photon path inside and outside the cloud complicates the situation. Due to photon penetration into the cloud, for given optical thickness and particle properties, the radiance measured by the EPIC A- and B-band sensors is not only a function of cloud top height, but also a function of cloud extinction coefficient profile. Except under special situations, e.g., for optically thick clouds over
dark surfaces with vertically uniform extinction coefficient (Yang et al., 2013), the EPIC measurements generally do not provide enough information content to retrieve the actual cloud top, but they are sufficient for retrieving another important cloud location information – namely CEP/CEH, as well as the effective cloud fraction (ECF).

CEP is equivalent to the mean pressure from which light is scattered and ECF is the derived radiometrically equivalent cloud fraction assuming an *a priori* cloud albedo (Stammes et al, 2008). CEP and ECF are important information and both
have been widely applied to trace gas retrievals and climate studies (e.g., Joiner et al., 2012, Wang et al. 2011, Vasilkov et al. 2008, Stammes et al. 2008, Sneep et al. 2008). CEP and ECF are retrieved using the Mixed Lambertian-Equivalent Reflectivity (MLER) concept, which has been extensively studied and applied to operational settings (e.g. Joiner et al., 2012, Yang et al. 2013, Koelemeijer et al., 2001). The MLER model assumes that the pixel contains two Lambertian reflectors, the surface and the cloud. The cloud is assumed opaque; hence no photon transmission occurs. The reflectance observed at the
sensor can be expressed as:

$$R_{abs}(\theta, \theta_0) = (1 - A_c)\alpha_s T_{abs}(P_s, \theta, \theta_0) + A_c \alpha_c T_{abs}(P_c, \theta, \theta_0) \tag{4}$$

$$R_{ref}(\theta, \theta_0) = (1 - A_c)\alpha_s T_{ref}(P_s, \theta, \theta_0) + A_c \alpha_c T_{ref}(P_c, \theta, \theta_0) \tag{5}$$

where $R_{abs}$ and $R_{ref}$ are the observed reflectances for the absorption and the reference channels, respectively, $A_c$ is the ECF, $\alpha_s$ is the surface albedo, $\alpha_c$ is the *a priori* cloud albedo, $T_{abs}$ and $T_{ref}$ are the two-way atmospheric transmittances for the
absorption and the reference channels, respectively, $P_c$ and $P_s$ are the CEP and the surface pressure, respectively, and $\theta$ and $\theta_0$ are the view zenith and solar zenith angles, respectively. Generally, $\alpha_c$ is assumed to be 0.8, which corresponds to an optically thick cloud. This value is selected to justify the no transmission assumption of the MLER model. It has been demonstrated that although the selection of the *a priori* cloud albedo does affect the results of the ECF, its effect on the retrieved CEP is relatively small (Stammes et al. 2008, Koelemeijer et al., 2001).

CEP and ECF can be retrieved with Eq. (4) and (5) through iteration, after which CEP is converted to cloud effective height (and, in the COT retrieval, cloud effective temperature; see section 2.3) using co-located atmospheric profiles



provided by the Goddard Earth Observing System Model, Version 5 (GEOS-5) (Lucchesi, 2015). The retrieval of CEP only requires one pair of absorption and reference channel observations; hence two independent EPIC CEPs can be obtained using the A-band and B-band pairs. In theory, the two CEPs are very close to each other if the surface albedo is the same for the two bands and instrument noise is not considered. There exists a subtle difference between the two CEPs, however,

because of the difference in photon penetration depths, which contains information on cloud vertical structure (Yang et al. 2013). Due to weaker absorption, photons in the B-band penetrate deeper into the cloud and result in a slightly higher CEP (lower altitude). As a result, both the A- and B-band CEPs are reported in the EPIC Cloud Product for the community to explore. For applications that need specific cloud effective height, such as trace gas retrieval correction, the A-band value is recommended as it is less noisy. Figure 4a, b shows the two CEP retrievals for the same granule as in Figure 2. As can be

seen from the scatter plot in Figure 4c, the A- and B-band CEPs are generally close to each other, with the B-band CEP higher. From radiative transfer simulations, the differences between the two CEPs resulting from penetration depth difference should be small (Yang et al. 2013). In addition, potential instrument instability, surface albedo differences, calibration accuracy and changes resulting from the differences in observation time (see Fig. 1) can also cause differences in the two CEPs.

**2.3 EPIC COT Algorithm**

Simultaneously retrieving COT and cloud effective radius (CER) using a combination of absorbing and non-absorbing spectral channels has been common practice (e.g., Platnick et al. 2017, Nakajima and King, 1990). However, due to the lack of a particle size-sensitive absorbing shortwave/mid-wave infrared channel, the EPIC COT retrieval adopts a single channel approach similar to what has been used by the International Satellite Cloud Climatology Project (ISCCP) (Rossow and

Schiffer, 1999) and the Multi-angle Imaging SpectroRadiometer (MISR) mission (Marchand et al. 2010). The 780 nm and 680 nm channels are used for retrievals over ocean and over land, respectively. Fixed particle sizes are assumed based on the MODIS global cloud effective radius modes derived from the Collection 6 (C6) MODIS cloud products (14 μm for liquid clouds and 30 μm for ice clouds). Using MODIS data, it has been shown that the uncertainties for a single channel retrieval due to assuming a fixed cloud effective radius are roughly 10% for liquid clouds and 2% for ice clouds (Meyer et al. 2016).

The EPIC COT algorithm shares the same code base, forward model assumptions and, to the extent possible, ancillary usage as the current C6/C6.1 MODIS cloud optical/microphysical property retrievals (MOD06) (Platnick et al., 2017). Since EPIC does not provide enough information to confidently determine cloud thermodynamic phase, two COT values are retrieved and reported for each cloudy pixel by assuming liquid and ice phases, respectively. Nevertheless, by combining the A- and B-band CEPs and GEOS-5 atmospheric profiles, the EPIC cloud product also provides A- and B-band cloud effective

temperatures as well as a most likely cloud thermodynamic phase derived from thresholds applied to the A-band CEP and temperature. Providing two COT values is useful to the community for further synergistic research, when more information





on cloud phase is available from other sources. An example of EPIC COT and the most likely cloud phase product is shown in Figure 5 for the same granule as in Figure 2.

## 3 Performance Assessment with Co-located GEO/LEO Results

An initial performance assessment has been conducted by comparing the EPIC cloud products with co-located cloud retrievals from GEO/LEO satellites. A GEO/LEO composite dataset has been generated by the Clouds and the Earth's Radiant Energy System (CERES) team at the NASA Langley Research Center by projecting the GEO/LEO retrievals to the EPIC grid (Khlopenkov et al., 2017). This composite dataset is produced in a two-step process to convert LEO and GEO data into an EPIC-view perspective. An algorithm for optimal merging of selected radiances and cloud properties derived from multiple satellite imagers produces nearly seamless global composites at a fixed 5-km resolution at each EPIC observation time. The composite data are subsequently remapped into the EPIC-view domain by convolving composite pixels with the EPIC point spread function (PSF) defined with a half-pixel accuracy. PSF-weighted average radiances and cloud properties are computed separately for each cloud phase. The merging process uses the GEO/LEO measurements nearest to the EPIC observation time to create the composite. The GEO platforms used include the Geostationary Operational Environmental Satellites (GOES), the MeteoSat satellites, the Multifunctional Transport Satellites (MTSAT) and the Himawari-8 satellites. The LEO sensors used include MODIS on NASA's Terra and Aqua, and AVHRR on NOAA's satellites. The retrieval and projection methods used are described in Minnis et al. (2011) and Khlopenkov et al. (2017). We note that the retrievals from the GEO/LEO platforms have uncertainties as well, but this dataset serves as an independent source for the assessment and consistency check of the EPIC cloud products.

## 3.1 Performance of EPIC Cloud Mask

Since the data from the GEO/LEO satellite instruments have finer spatial resolutions than EPIC, each EPIC pixel is likely to contain many GEO/LEO instrument pixels; hence a sub-pixel cloud fraction can be calculated for each co-located EPIC pixel. Figure 6 shows an example. Comparing the EPIC RGB image (Figure 6a) with the corresponding cloud mask (Figure 6b), it can be seen that the EPIC cloud mask performs well for this case. Figure 6c is the corresponding sub-pixel cloud fraction for each EPIC pixel from the GEO/LEO composite. From the cumulative distribution of the GEO/LEO composite in Figure 6d, 37% of the EPIC pixels are fully cloudy and 16% are fully clear; the remainder (84%) of the EPIC pixels have some level of partial cloudiness. The EPIC cloud mask gives a cloud fraction for the entire scene of 66.4%. While we re-emphasize that a portion of the EPIC pixels are only partially cloudy, the comparison between the global cloud coverage derived from EPIC and from the GEO/LEO composite is still meaningful in understanding the performance of the EPIC cloud mask.

Figure 7 compares the global average cloud fraction from EPIC and from the GEO/LEO composites from four weeks of data, one from each season, in 2016 (Mar. 6-12, Jun. 20-26, Sept. 21-27, and Dec. 20-26). For this study, the global statistics



are done with a customized version of the composites that uses only GEO/LEO data within +/- 5 minutes of the EPIC observation time to minimize the temporal differences between the EPIC and GEO/LEO data. As shown in the figure, the two datasets visually match each other well. The total global cloud fractions are 70.7% and 71.8% for EPIC and the GEO/LEO composites, respectively (Table 3). Over land it is 60.9% for EPIC and 57.9% for GEO/LEO; over ocean it is

74.7% for EPIC and 77.5% for GEO/LEO. Discrepancies exist between the two datasets; the most obvious one is over snow- and ice-covered regions. For example, EPIC misses a large portion of clouds over West Antarctica, where cloud detection uses the $O_2$ A- and B-band ratios. Ongoing efforts are taking place to improve cloud masking performance over these regions.

## 3.2 Performance of EPIC CEP/CEH

As discussed above, the retrieved CEPs are generally higher (lower altitude) than the physical cloud top pressure due to the photon penetration into clouds that is ignored in the adopted MLER concept. However, at the most oblique solar/view zenith angles, CEPs can be lower (higher altitude) than the physical cloud top due to the contribution of Rayleigh scattering (e.g., Yang et al. 2013, Ferlay et al. 2010, Vanbauce et al., 2003).

Figure 8 compares the EPIC A-band CEP with the GEO/LEO composites. Cloud pressures in the GEO/LEO composite

are derived from infrared channels, which are radiative pressures but are closer to the physical cloud top. As shown in the figure, the EPIC A-band CEPs are generally higher (lower altitude) than those of the GEO/LEO composites (note that the vertical axis is reversed), except at the edges of the image where the solar and view zenith angles are both large. The mean cloud pressures along this line are 658 mb for the EPIC A-band and 645 mb for the GEO/LEO composite. These results show that the EPIC CEPs are consistent with previous studies and theoretical predictions.

Figure 9a compares the EPIC A- and B-band CEP retrievals using the same four weeks of data as in Figure 7. Results are similar to the single granule case shown in Figure 4c. The A- and B-band CEPs are generally close to each other with the B-band CEP higher (lower altitude). The spread of the scatter plot can come from spectral surface albedo differences, changes resulting from the differences in observation time, calibration accuracy etc. Figure 9b and 9c show the comparison between EPIC A- and B-band CEP and cloud pressure from GEO/LEO composites. In general, the GEO/LEO cloud pressure

are lower (higher in altitude) than the CEPs, as their sensitivity lies is closer to the physical cloud top. Again, the results are consistent with existing studies and theoretical expectations.

## 3.3 Performance of EPIC COT Retrieval

As mentioned above, the EPIC COT retrieval adopts a single channel approach where the 780 nm and 680 nm channels are used for retrievals over ocean and over land, respectively, and the algorithm shares the same code base, forward model

assumptions, and ancillary usage as the current MODIS C6/C6.1 cloud optical property products. A separate comprehensive study has been conducted to investigate the feasibility and uncertainty of the EPIC COT retrieval algorithm (Meyer et al.


2016). The study showed that for ice clouds, uncertainties are mostly less than 2%, because even though a fixed particle size is assumed (30μm), the ice crystal model used in the retrieval (i.e., severely roughened aggregate of hexagonal columns) (Yang et al., 2013a; Holz et al., 2016) is not sensitive to the particle size; for liquid clouds the uncertainty is larger, roughly 10%, although for thin clouds (COT < 2) the error can be higher.

Figure 10 shows a comparison of the two EPIC COT retrievals (liquid and ice phase) with the GEO/LEO composites. In general, there is good correlation between the EPIC and GEO/LEO composite COTs. For reference, the mean EPIC COT along the selected line is 8.9 for liquid phase, 7.4 for ice phase, and 9.5 for the GEO/LEO composite. Using the same four weeks of data as in Figure 7, Figure 11a shows the relationship between the two EPIC COT retrievals by assuming ice and liquid phase, respectively. As expected, for the same reflectance, liquid clouds have higher COT than ice clouds  (King et al.,
2004). Figure 11b and 11c compares the EPIC COT with the GEO/LEO COT for liquid and ice phase clouds, as identified by the GEO/LEO retrieval algorithm (Minnis et al., 2011). In general, the two products match each other well. We emphasize again that the COT retrievals used in the GEO/LEO composites are generated by the CERES-Cloud team with a different set of assumptions and cloud models (Minnis et al., 2011) than are used in the EPIC COT retrievals. Other factors that can contribute to the spread include the differences in observation time, the instrument spatial resolution, etc.

## 4 Summary and Discussion

     From the Earth's L1 Lagrangian point, EPIC has been providing a continuous view of the sunlit side of the Earth since June 2015. Observations from the 10 EPIC spectral channels are a unique dataset for cloud system monitoring and cloud product development. A suite of algorithms has been developed to generate the standard EPIC Level 2 Cloud Products that include cloud mask, CEP/CEH, COT, etc. These products are archived at the Atmospheric Science Data Center at the NASA
Langley Research Center.

     The EPIC cloud product algorithms are presented. The EPIC cloud mask adopts the threshold method and utilizes the BRFs of the 388, 680, and 780 nm channels, the 764/780 nm (A-band) ratio, and the 688/680 nm (B-band) ratio as tests. The Earth's surface is separated into three types: land, ocean, and snow/ice; two tests are applied to each surface type and the results from each test are combined to generate the final cloud mask, which classifies a pixel as clear or cloudy with
confidence levels. The EPIC CEPs are derived with observations from the $O_2$ A-band (780 nm and 764 nm), and B-band (680 nm and 688 nm) pairs based on the MLER model. Both A-band CEP and B-band CEP are reported in the cloud product. CEPs are converted to cloud heights using the co-located atmospheric profiles provided by the GEOS-5 model. Due to the lack of particle size sensitive channels, the EPIC COT retrieval adopts a single channel approach where a fixed particle size is assumed. Observations from the 780 nm and 680 nm channels are used for retrievals over ocean and over
land, respectively. In addition, since the EPIC channels do not contain enough information to confidently determine cloud thermodynamic phase, the EPIC COT product provides two independent retrievals for each cloudy pixel, one assuming





liquid phase and one assuming ice phases, respectively. A most likely cloud thermodynamic phase is also provided based on thresholds applied to the CEP and the cloud effective temperature derived from the EPIC $O_2$ A-band.

An initial comparison with co-located GEO/LEO results shows that the EPIC cloud products are performing well. Based on the analysis of four weeks of data from the year 2016, the total global cloud fractions are 70.7% and 71.8% for EPIC and the GEO/LEO composites, respectively. Due to photon penetration, the EPIC CEPs are generally higher (lower altitude) compared to the GEO/LEO retrievals from IR channels that have sensitivity closer to the physical cloud top. The EPIC CEP retrievals are consistent with theoretical expectations. The EPIC COT retrieval shares the same code base, forward model assumptions, and ancillary usage as the current C6/C6.1 MODIS cloud optical/microphysical property retrievals (MOD06).

The EPIC Cloud Products provide cloud properties of the sunlit side of Earth for climate studies and for generating other geophysical products that require cloud properties as input. Known issues include the cloud detection problems over ice and snow, which leads to errors in CEP/CEH and COT retrievals. Ongoing efforts are taking place to improve the EPIC Cloud Products in these regions.

**Acknowledgments**

This research was supported by the NASA DSCOVR Earth Science Algorithms program managed by Richard Eckman. The DSCOVR level-1 and level-2 data used in this paper are publicly available from NASA Langley Atmospheric Sciences Data Center (ASDC).

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



**Table 1.** List of current EPIC Cloud Products and required EPIC observations. The native EPIC pixel size is roughly 8 km at nadir.

| Product | Resolution | Description | EPIC Data Used |
|---------|------------|-------------|----------------|
| Cloud Mask | Native pixel size | Each EPIC pixel is classified as clear/cloudy with high/low confidence. | 388 nm, 680 nm, and the 780 nm relectances, the 764/780 nm and the 688/680 nm ratios. |
| CEP/CEH | Native pixel size | CEP/CEH is retrieved for both $O_2$ A- and B-bands | $O_2$ A-band (780 nm and 764 nm), and B-band (680 nm and 688 nm), EPIC cloud mask |
| COT | Native pixel size | Cloud optical thickness is retrieved using a single-channel approach (680 nm over land and 780 nm over ocean) | 680 nm (over land), 780 nm (over ocean), EPIC cloud mask, EPIC CEP |

**Table 2.** Cloud masking tests for different surface types.

| Test | Ocean | Land | Snow/Ice |
|------|-------|------|----------|
| 388 nm | | Y | |
| 680 nm | Y | | |
| 780 nm | Y | | |
| Ratio: 688 nm / 680nm | | | Y |
| Ratio: 764 nm / 780 nm | | Y | Y |





**Table 3.** Comparison between EPIC cloud fraction and collocated GEO/LEO results. Data used are the same as in Figure 7.

| Surface Type | EPIC | Collocated GEO/LEO |
|---|---|---|
| Land | 60.9% | 57.9% |
| Ocean | 74.7% | 77.5% |
| Ice and Snow | 69.0% | 75.2% |
| Global | 70.7% | 71.8% |

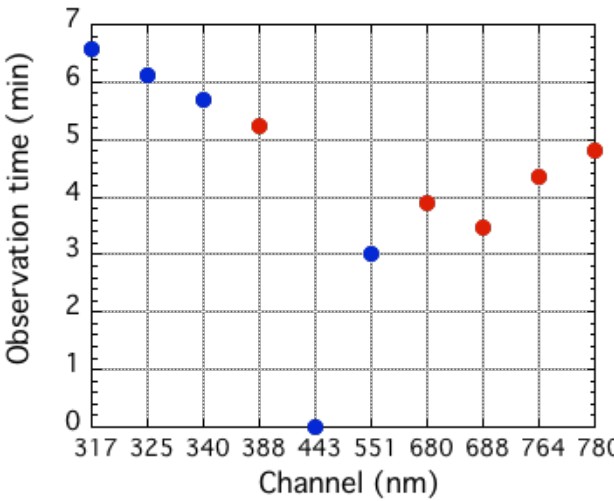

**Figure 1.** Observation time from the start of acquisition for each EPIC channel starting from 443 nm. Red dots are the channels used for cloud retrievals. It takes about 7 min to complete the 10-channel image set.

25



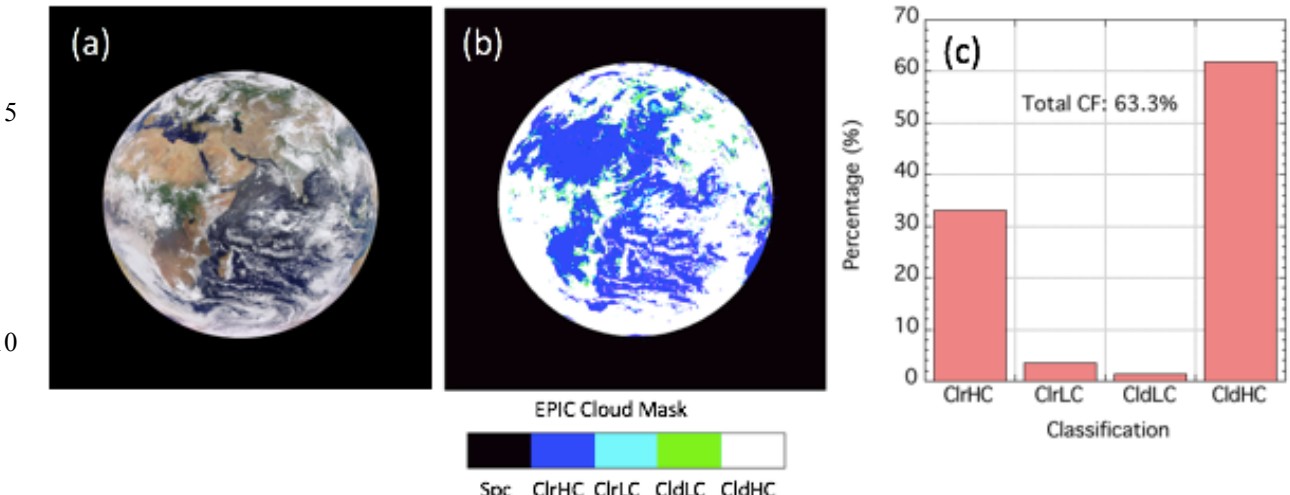

**Figure 2.** Sample EPIC L2 cloud products for the observations at 08 UTC on Aug. 18, 2016: (a) EPIC RGB image; (b) EPIC cloud mask. Spc: space pixels, ClrHC: high confidence clear, ClrLC: low confidence clear, CldLC: low confidence cloudy, and CldHC: high confidence cloudy; (c) percentage of each scene type derived from the cloud mask.

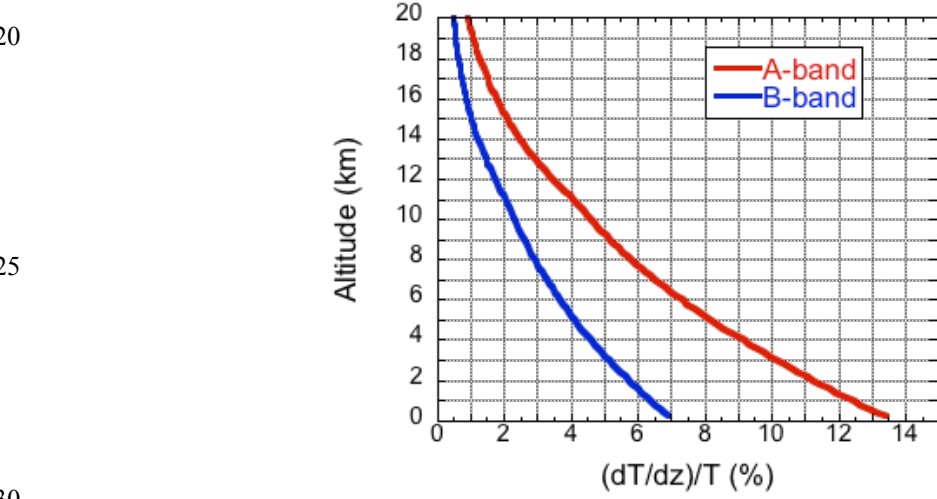

**Figure 3.** Derivative of the clear sky two-way transmittance (dT/dz) for the EPIC A- and B-band absorption channels normalized by T, where T is the two-way transmittance and z is the height. The plot shows how much T changes (%) if the altitude of the reflecting layer changes by 1 km.





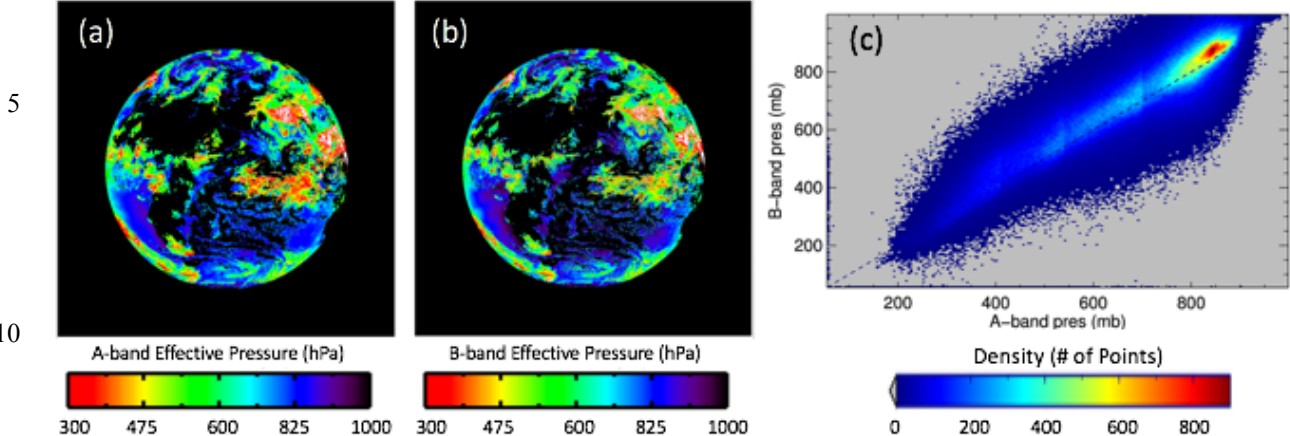

**Figure 4:** Sample EPIC L2 cloud effective pressure (CEP) products. Same granule as Figure 3, but for (a) $O_2$ A-band CEP, (b) $O_2$ B-band CEP, and (c) scatter plot for A- vs B-band CEP.

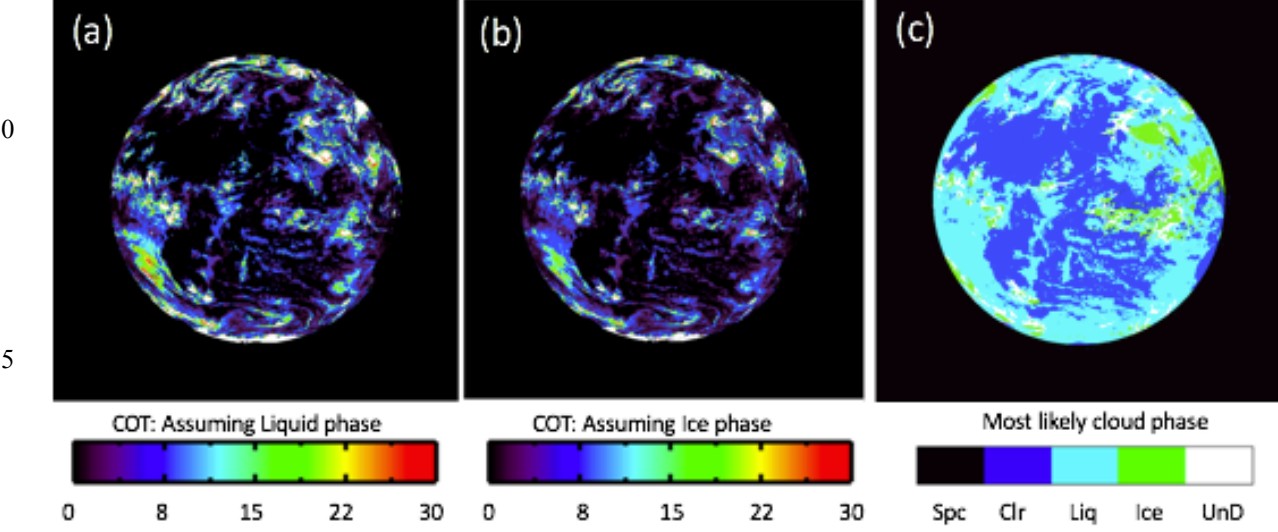

**Figure 5:** Sample EPIC L2 cloud optical thickness (CODT) products. Same granule as Figure 3, but for (a) COT retrieval assuming liquid thermodynamic phase, (b) COT retrieval assuming ice thermodynamic phase, and (c) the most likely cloud phase based on cloud effective temperature derived from the A-band CEP.



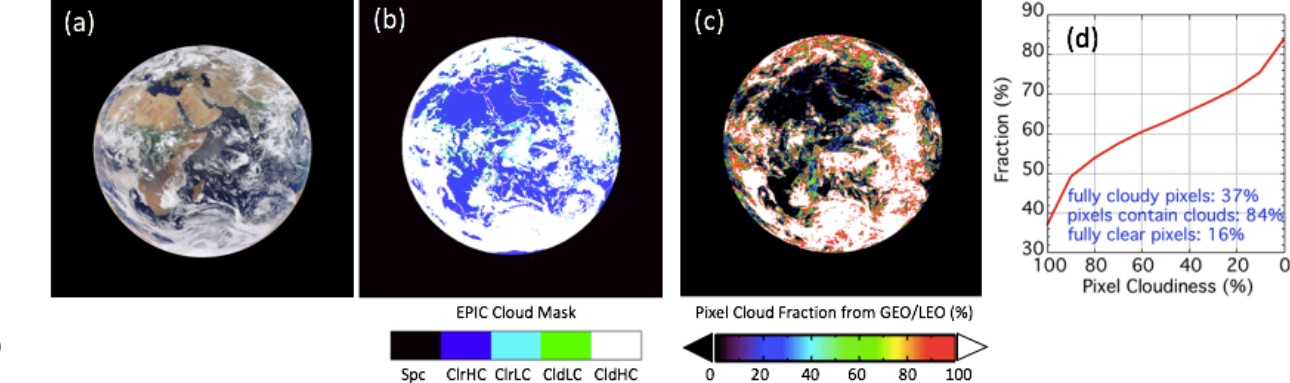

**Figure 6.** Comparison of the EPIC cloud mask with the co-located GEO/LEO satellite results. (a) The EPIC RGB image for 08 UTC on Sept. 15, 2015; (b) the corresponding EPIC cloud mask; (c) the corresponding GEO/LEO composite for the sub-pixel cloud fraction of each EPIC pixel; (d) the cumulative distribution of pixels from 100% cloudy to fully clear for this granule based on the GEO/LEO composite in (c).

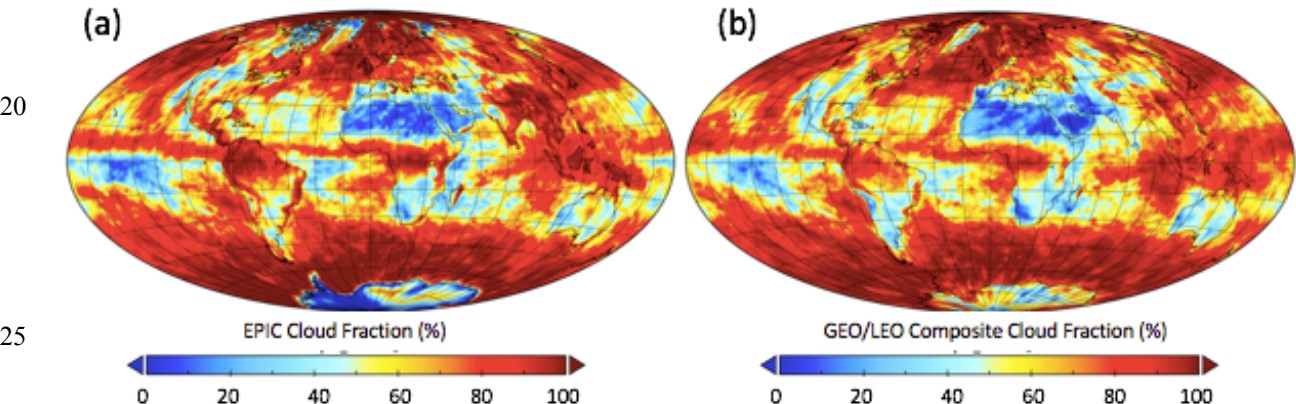

**Figure 7:** Comparison between EPIC global cloud coverage and co-located (spatially co-located and time difference within 5 min) GEO/LEO results. Four weeks of data from 2016 are utilized with one week from each season (Mar. 6-12, Jun. 20-26, Sept. 21-27, and Dec. 20-26). (a) Average EPIC cloud coverage; and (b) average cloud coverage from the GEO/LEO composites.



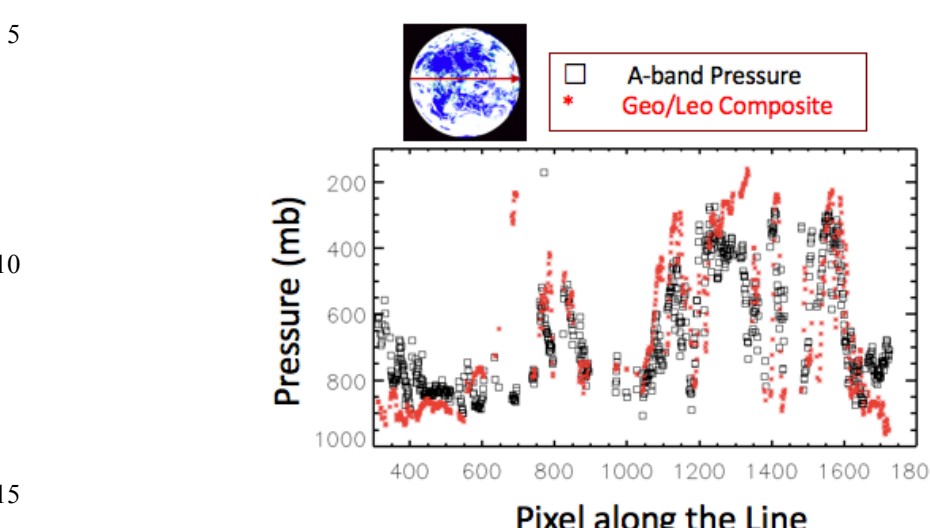

**Figure 8.** Comparison of EPIC A-band CEP with the co-located GEO/LEO retrievals for the same granule as Figure 6. The comparison is done along the red arrow shown on the cloud mask thumbnail at top.





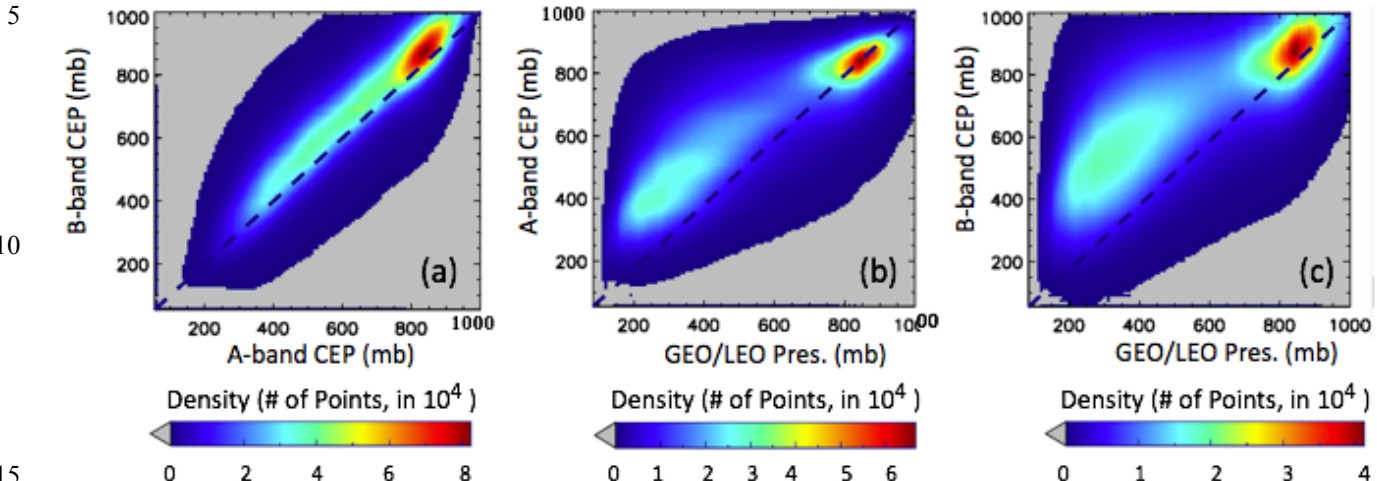

**Figure 9:** EPIC A- and B-band CEP and the comparison with cloud pressure from GEO/LEO composites. Data used are the same as in Figure 7. Scatterplots (a) for EPIC A- vs. B-band CEP; (b) the GEO/LEO composites cloud pressure vs EPIC A-band CEP; and (c) the GEO/LEO composites cloud pressure vs EPIC B-band CEP.



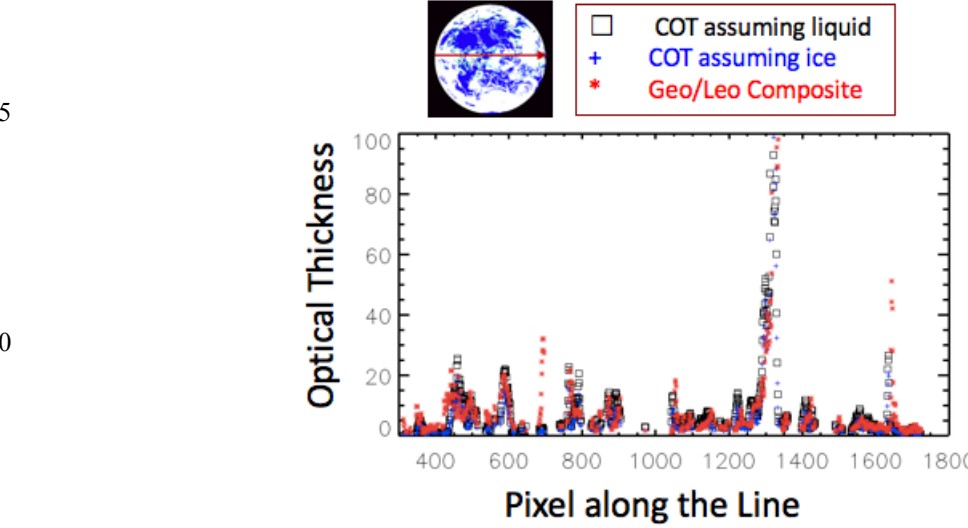

15 **Figure 10.** Comparison of EPIC COT with the collocated GEO/LEO retrievals for the same granule as in Figure 6. The comparison is done along the red arrow shown on the cloud mask thumbnail at top.

25

30





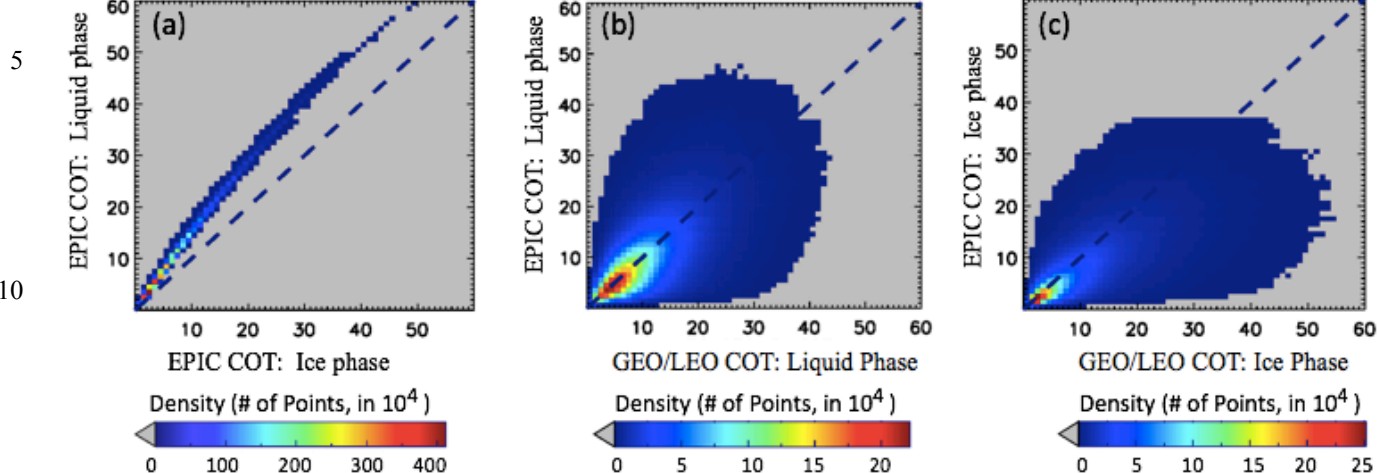

**Figure 11:** (a) Scatterplot for EPIC COT assuming liquid vs ice phase; (b) EPIC COT vs GEO/LEO composites: liquid clouds only as identified by the GEO/LEO algorithms; and (c) EPIC COT vs GEO/LEO composites: ice clouds only as identified by the GEO/LEO algorithms. Data used are the same as in Figure 7.