# Peer review of "Cloud Products from the Earth Polychromatic Imaging Camera (EPIC): Algorithms and Initial Evaluation"

_Atmospheric Measurement Techniques, 2018_

## Referee Comment (RC1) · Anonymous Referee #1 · 28 Nov 2018

This manuscript presents a simple algorithm description for deriving cloud properties from the EPIC camera located on the DSCOVR satellite, and makes initial comparisons to independent observations derived from combinations of LEO/GEO data. The cloud properties reported include a cloud mask, cloud top height (CTH) & temperature (CTT), effective cloud fraction (ECF), and cloud optical thickness (COT). In the case of COT, both liquid and ice phase COT retrievals are reported. A total of four weeks of data are investigated, with one week drawn from each of the seasons.

EPIC is making a first of a kind observation from Lagrange point 1 and this study certainly warrants publication. This is a short and concise paper that is well organized

and to the point. The only problem is a lack of background description on how the data is obtained in the instrument and re-gridded and then used for Level 2 cloud retrievals. If this has been described at length in previous papers or ATBDs, this information needs to be conveyed in this paper because it is not clear how this unique viewing perspective of Earth and 7-minute long data acquisition cycle are reconciled.

To be specific, the Earth rotates 15 degrees in an hour, which translates to 1.75 degrees in 7 minutes. If the channels are acquired in sequence (is this the case? And If so, does it start with 317 nm and end with 780 nm?), the Earth advances 0.175 degrees in 0.7 minutes (42 seconds) during one channel acquisition, which is about 19.4 km of Earth rotation at the equator. Therefore, from channel to channel, the Earth rotates about 2.5 pixels in distance at nadir. How is this dealt with? The authors need to have some added discussion on this, or clarify that this is incorrect and that the instrument operates in a different manner.

Table 1 states that the cloud products are obtained at the "native pixel size", which is about 8 km at nadir. How much bigger do the pixels get at the edge of the Earth disk? And how is the "re-gridding" consistent with keeping a native 8 km resolution at nadir?

Other comments

Throughout the manuscript "etc." is used in place of quantitative information. These instances must spell out exactly what is intended by "etc." because in all cases it is not clear at all what is meant.

p. 1 line 27: . . .capability not previously available. . . p. 1 line 28: delete "in the past" p. 2 line 24: . . .retrieval are run. . . p. 4 line 2: what is the impact of the 6 m/s assumption? Can the authors add an error estimate(s) based on known climatologies of wind speeds within and outside of storms in different latitude bands? p. 4 line 4: relatively p. 7 lines 10-11: what is the size of the EPIC pixel PSF at half-max? Are they the same for all 10 channels and all elements of the CCD array?

---

## Referee Comment (RC2) · Anonymous Referee #3 · 30 Dec 2018

This paper provides a brief and succinct description of the algorithms leading to the cloud products from the EPIC instrument on the DISCOVR spacecraft. The results are compared with similar retrievals from geostationary and polar-orbiting satellite instruments. I have only minor comments regarding the conclusions drawn from the comparison with other retrievals. The paper should be suitable for publication after some minor revision.

In the abstract (line 21) and again in section 3.2 (line 9, p. 8) the authors claim that the comparison of the EPIC retrievals with retrievals from other instruments demonstrate that the EPIC retrievals are "consistent with theoretical expectations" or "theoretical

predictions". But these claims are not clearly justified. Can the authors elaborate on what they mean by "theoretical expectations" and clarify quantitatively how the EPIC results demonstrate consistency?

Furthermore, the comparisons are not discussed in any sort of quantitative manner in the narrative. While the quantitative comparison is present in the figures, the text provides merely qualitative conclusions such as "in general, the two products match each other well" (line 32, p.8). This, of course is a close to meaningless statement when comparing two quantities that each have some uncertainty. Much more meaningful would be if they agree within the range of expected uncertainty. And if that is the case, then naturally one would need to know the reasonable range of uncertainty for the retrievals. If the authors expect other members of the community to use these products and cite this paper as evidence that they are suitable for atmospheric research purposes, then they should make a credible effort to offer realistic uncertainty bounds.

---

## Author Response (AR1)

[revised manuscript text omitted]

**RC1 comments:**

*This manuscript presents a simple algorithm description for deriving cloud properties from the EPIC camera located on the DSCOVR satellite, and makes initial comparisons to independent observations derived from combinations of LEO/GEO data. The cloud properties reported include a cloud mask, cloud top height (CTH) & temperature (CTT), effective cloud fraction (ECF), and cloud optical thickness (COT). In the case of COT, both liquid and ice phase COT retrievals are reported. A total of four weeks of data are investigated, with one week drawn from each of the seasons.*

*EPIC is making a first of a kind observation from Lagrange point 1 and this study certainly warrants publication. This is a short and concise paper that is well organized and to the point. The only problem is a lack of background description on how the data is obtained in the instrument and re-gridded and then used for Level 2 cloud retrievals. If this has been described at length in previous papers or ATBDs, this information needs to be conveyed in this paper because it is not clear how this unique viewing perspective of Earth and 7-minute long data acquisition cycle are reconciled.*

*To be specific, the Earth rotates 15 degrees in an hour, which translates to 1.75 degrees in 7 minutes. If the channels are acquired in sequence (is this the case? And If so, does it start with 317 nm and end with 780 nm?), the Earth advances 0.175 degrees in 0.7 minutes (42 seconds) during one channel acquisition, which is about 19.4 km of Earth rotation at the equator. Therefore, from channel to channel, the Earth rotates about 2.5 pixels in distance at nadir. How is this dealt with? The authors need to have some added discussion on this, or clarify that this is incorrect and that the instrument operates in a different manner.*

**Response:**

We thank the reviewer for the insightful comments. Rotation of the Earth is indeed an issue that the EPIC L1B algorithm has to take into account when projecting all the different channels onto a common grid. Even after the reprojection, the problem still exists for cloud retrievals, because of the potential cloud field change due to the latency between imaging different wavelengths. As mentioned in the manuscript, we selected the five channels that are close to each other in imaging time in order to mitigate the problem. We added the following paragraphs to address the issues raised by the reviewer:

"The focal plane of the EPIC system is a 2048 x 2048 pixel CCD array. The point spread function of the CCD array has a full width at half maximum (FWHM) of ~1.34 pixels. Images for the 10 spectral channels are obtained using a 10-color filter wheel assembly and shutter, the operation of which takes about 7 minutes to acquire a complete set of observations. As shown in Figure 1, the observation starts with the 443nm channel, followed by the 551nm, 688nm, 680nm, 764nm, 780nm, 388nm, 340nm, 325nm, and 317nm channels. The time difference between one channel and the next consists of readout, exposure, and filter rotation time. Limited by data transmission capability, only the 443 nm channel image is downlinked at its the original size; the rest are all reduced to 1024 x 1024 pixels through onboard processing and then interpolated back to the full size of 2048 x 2048 after being downlinked. As can be seen from Figure 1, the observation time difference between one channel and the next is usually half a minute, except between the 443nm and 551nm, which is ~3minutes. At full resolution, the pixel size of EPIC observations is ~8km at nadir. For view zenith angle (VZA) > 0, the EPIC pixels become elliptical, where the longer axis increases by a factor of about 1/cos(VZA) and the shorter axis remains ~8km.

Due to the latency between imaging different wavelengths and the rotation of the Earth, the regions covered by the images of the 10 EPIC spectral channels are not exactly the same. For algorithm development and research, all 10 spectral channels are projected to a common grid in the Level-1B (L1B) radiance product. The projection procedure includes (see details in Marshak et al. 2018): 1) mapping the images to a 3D model of the Earth in order to calculate the geolocation of each pixel; and 2) projecting and regridding each image onto the common reference grid. "

*Table 1 states that the cloud products are obtained at the "native pixel size", which is about 8 km at nadir. How much bigger do the pixels get at the edge of the Earth disk? And how is the "re-gridding" consistent with keeping a native 8 km resolution at nadir?*

**Response:**

Thanks for pointing this out. As mentioned in our answer to the previous question, At full resolution, the pixel size of EPIC observations is ~8km at nadir. For view zenith angle (VZA) > 0, the EPIC pixels become elliptical, where the longer axis increases by a factor of about 1/cos(VZA) and the shorter axis remains ~8km. We tried to separate the term "resolution" and "pixel size", because after the onboard 2x2 averaging for the nine channels other than 443 nm, the resolution becomes ~18km at nadir.

**Other comments**

*Throughout the manuscript "etc." is used in place of quantitative information. These instances must spell out exactly what is intended by "etc." because in all cases it is not clear at all what is meant.*

**Response:**

Changes are made to address this issue. Occurrences of "etc." are removed for the following places, where we think the possibilities are enumerated: P1, Line 15; P2, Line 25; and P10, Line 5. Other occurrences of "etc." are kept, because of unknown or too many possibilities, such as the number of potential applications of cloud products.

*p. 1 line 27: . . .capability not previously available. . .*

**Response:** done

*p. 1 line 28: delete "in the past"*

**Response:** done

*p. 2 line 24: . . .retrieval are run. . .*

**Response:** done

*p. 4 line 2: what is the impact of the 6 m/s assumption? Can the authors add an error estimate(s) based on known climatologies of wind speeds within and outside of storms in different latitude bands?*

**Response:**

This is an error on our part. For the current version, the thresholds over ocean surfaces are actually empirically derived. We added text to make this clear. Now the paragraph reads as follows:

  "Similar to the 388 nm test over land, a Rayleigh correction is also applied to both the 680 nm and 780 nm BRFs. Thresholds are derived empirically. BRF values $T_{680} = 0.11$ and $T_{780} = 0.10$ are used to separate low confidence clear and low confidence cloudy scenes for the 680nm and 670nm channels, respectively. $T_{680} \pm 0.03$ and $T_{780} \pm 0.03$ are used as the high confidence thresholds for the two channels, respectively."

*p. 4 line 4: relatively p. 7 lines 10-11: what is the size of the EPIC pixel PSF at half-max? Are they the same for all 10 channels and all elements of the CCD array?*

**Response:**

The size of the EPIC pixel PSF is about 1.34 pixels and is the same for all the ten channels. Text has been added to the manuscript to address this issue.

**Response:**

We thank the reviewer for the comments. We have added the following text to the manuscript to clarify the issue: "The EPIC cloud pressure is essentially the centroid of the reflected photons registered at the satellite sensor. Photons penetrating into and through clouds have longer path lengths compared to photons reflected at the cloud top. Since the MLER model does not take these factors into account, it is expected that the EPIC CEP is lower in altitude than the physical cloud top (higher in pressure); hence the results shown here are consistent with previous studies and theoretical predictions."

We also added Figure 3a to better illustrate the MLER model:

[Figure]

Figure 3b: the two types of photon paths considered in the MLER method. The picture shows one partially cloudy EPIC pixel.

*Furthermore, the comparisons are not discussed in any sort of quantitative manner in the narrative. While the quantitative comparison is present in the figures, the text provides merely qualitative conclusions such as "in general, the two products match each other well" (line 32, p.8). This, of course is a close to meaningless statement when comparing two quantities that each have some uncertainty. Much more meaningful would be if they agree within the range of expected uncertainty. And if that is the case, then naturally one would need to know the reasonable range of uncertainty for the retrievals. If the authors expect other members of the community to use these products and cite this paper as evidence that they are suitable for atmospheric research purposes, then they should make a credible effort to offer realistic uncertainty bounds.*

**Response**:

We agree with the reviewer. We redid analysis attempting to provide uncertainty estimates. The following text and results are given in Section 3 of the revised manuscript:

[revised manuscript text omitted]

**SC1 comments:**

*I have three comments to this interesting paper:*

*- page 2, section 2.1: the threshold values used for the different cloud mask tests should be specified*

- **Response**:

We thank Dr. Loyola for his interest in this paper and for the comments.

We made changes to the threshold description over ocean surface and added the threshold values to the text: "Thresholds are derived empirically. BRF values $T_{680} = 0.11$ and $T_{780} = 0.10$ are used to separate low confidence clear and low confidence cloudy scenes for the 680nm and 680nm channels, respectively. $T_{680} \pm 0.03$ and $T_{780} \pm 0.03$ are used as the high confidence thresholds for the two channels, respectively."

As mentioned in the text, over land and ice and snow covered areas, the thresholds are a dataset that is a functions of surface height and surface reflectivity as described in Tilstra et al. (2017). For these surface types, we described the derivations of the thresholds in the text, which are relatively simple steps.

*- page 4, section 2.2: O2 A-Band cloud retrievals are also operational for GOME, GOME-2 and TROPOMI/Sentinel-5 Precursor, see Loyola et al. 2007 and Loyola et al., 2018.*

- **Response**:

We added references to the Loyola et al. 2018 and the Schüssler, 2014 papers

*- page 5: The statement "the EPIC measurements generally do not provide enough information content to retrieve the actual cloud top, but they are sufficient for retrieving another important cloud location information – namely CEP/CEH" is not correct. The main reason for retrieving a CEH instead of the cloud top height is the usage of a Lambertian cloud model instead of a more realistic Mie scattering cloud model, see Schüssler et al., 2014 and Loyola et al., 2018.*

**Response**:

The EPIC instrument has two points of measurements in the oxygen absorption spectrum (764nm and 688nm). Under special situations, e.g., for optically thick clouds over dark surfaces with vertically uniform extinction coefficient, it is possible to retrieve cloud top and cloud

geometrical thickness simultaneously (Yang et al., 2013). However, we have established that, in general, EPIC's $O_2$ channels do not provide enough independent information for vertical extinction profile retrieval (Davis et al., 2018a,b). Consequently, unless the vertical extinction profile is known or can be retrieved by some other means, we won't be able to separate the cloud thickness effect from the cloud top effect (e.g. Joiner et al 2012). Hence, on this point, we are not in agreement with Dr. Loyola.

- **References:**

- Davis, A.B., Merlin, G., Cornet, C., C.-Labonnote, L., Riédi, J, Ferlay, N., Dubuisson, P., Min, Q., Yang, Y., and Marshak, A.: Cloud information content in EPIC/DSCOVR's oxygen A- and B-band channels: An optimal estimation approach, *J. Quant. Spectrosc. Rad. Transf.*, 216, 6-16, doi:10.1016/j.jqsrt.2018.05.007, 2018a.

- Davis, A.B., Ferlay, N., Libois, Q., Marshak, A., Yang, Y., and Min, Q.: Cloud information content in EPIC/DSCOVR's oxygen A- and B-band channels: A physics-based approach, *J. Quant. Spectrosc. Rad. Transf.*, 220 84-96, https://doi.org/10.1016/j.jqsrt.2018.09.006, 2018b.

- Joiner, J., Vasilkov, A. P., Gupta, P., Bhartia, P. K., Veefkind, P., Sneep, M., de Haan, J., Polonsky, I., and Spurr, R.: Fast simulators for satellite cloud optical centroid pressure retrievals; evaluation of OMI cloud retrievals. *Atmos. Meas. Tech.,* 5, no. 3 (2012): 529-545, 2012.

- Loyola, D. G., Gimeno García, S., Lutz, R., Argyrouli, A., Romahn, F., Spurr, R. J. D., Pedergnana, M., Doicu, A., Molina García, V., and Schüssler, O.: The opera- tional cloud retrieval algorithms from TROPOMI on board Sentinel-5 Precursor, Atmos. Meas. Tech., 11, 409-427, https://doi.org/10.5194/amt-11-409-2018, 2018.

- Schüssler, O., Loyola, D., Doicu, A., and Spurr, R.: Information Content in the Oxygen A-band for the Retrieval of Macrophysical Cloud Parameters, IEEE Transactions on Geoscience and Remote Sensing, 52, 3246–3255, 2014.

- Tilstra, L. G., Tuinder, O. N. E., Wang, P., and Stammes, P.: Surface reflectivity climatologies from UV to NIR determined from Earth observations by GOME-2 and SCIAMACHY, J. Geophys. Res.-Atmos., 122, 4084–4111, 2017.

- Yang, Y., Marshak, A., Mao, J., Lyapustin, A., and Herman, J.: A Method of Retrieving Cloud Top Height and Cloud Geometrical Thickness with Oxygen A and B bands for the Deep Space Climate Observatory (DSCOVR) Mission: Radiative Transfer Simulations. *J. Quant. Spectrosc. Radiat. Trans.* 122, 141-149, http://dx.doi.org/10.1016/j.jqsrt.2012.09.017, 2013.

-